Intergenerational conflicts may help explain parental absence effects on reproductive timing: a model of age at first birth in humans

Moya Cristina cristina.moya@lshtm.ac.uk
Sear Rebecca
Department of Population Health, London School of Hygiene and Tropical Medicine , London , United Kingdom
Wagner Jennifer
Electronic publication date: 2014 Aug 12
Publication date: 2014
Volume: 2
Electronic Location ID: e512
Received 2014 Mar 31; Accepted 2014 Jul 22
Copyright: © 2014 Moya and Sear
Copyright year: 2014
Copyright holder: Moya and Sear
License: This is an open access article distributed under the terms of the Creative Commons Attribution License, which permits unrestricted use, distribution, reproduction and adaptation in any medium and for any purpose provided that it is properly attributed. For attribution, the original author(s), title, publication source (PeerJ) and either DOI or URL of the article must be cited.
License URL: https://creativecommons.org/licenses/by/4.0/

Keywords: Cooperative breeding, Life history theory, Intergenerational conflict, Father absence, Helpers at the nest, Mother absence, Reproductive decision-making, Kin competition, Parental investment, Senescence

Funding: European Research Council Starting Grant 263760 Authors were funded by the European Research Council. The funders had no role in study design, data collection and analysis, decision to publish, or preparation of the manuscript.

==============================
Background. Parental absences in childhood are often associated with accelerated reproductive maturity in humans. These results are counterintuitive for evolutionary social scientists because reductions in parental investment should be detrimental for offspring, but earlier reproduction is generally associated with higher fitness. In this paper we discuss a neglected hypothesis that early reproduction is often associated with parental absence because it decreases the average relatedness of a developing child to her future siblings. Family members often help each other reproduce, meaning that parents and offspring may find themselves in competition over reproductive opportunities. In these intergenerational negotiations offspring will have less incentive to help the remaining parent rear future half-siblings relative to beginning reproduction themselves.

Method. We illustrate this “intergenerational conflict hypothesis” with a formal game-theoretic model.

Results. We show that when resources constrain reproductive opportunities within the family, parents will generally win reproductive conflicts with their offspring, i.e., they will produce more children of their own and therefore delay existing offsprings’ reproduction. This is due to the asymmetric relatedness between grandparents and grandchildren (r = .25), compared to siblings (r = 0.5), resulting in greater incentives for older siblings to help rear younger siblings than for grandparents to help rear grandchildren. However, if a parent loses or replaces their partner, the conflict between the parent and offspring becomes symmetric since half siblings are as related to one another as grandparents are to grandchildren. This means that the offspring stand to gain more from earlier reproduction when their remaining parent would produce half, rather than full, siblings. We further show that if parents senesce in a way that decreases the quality of their infant relative to their offspring’s infant, the intergenerational conflict can shift to favor the younger generation.

Introduction

Many social scientists have shown that children who experience parental absences due to divorce or death consistently have earlier ages of puberty and first reproduction in post-industrial societies (Surbey, 1990; Nettle, Coall & Dickins, 2011; Ellis et al., 2003). This correlation is also seen in some developing societies (Sheppard, Snopkowski & Sear, 2014; Birdthistle et al., 2008) although the effects are less consistent in these contexts (Waynforth, Hurtado & Hill, 1998; Allal et al., 2004; Palermo & Peterman, 2009; Leonetti & Nath, 2009; Shenk et al., 2013; Winking, Gurven & Kaplan, 2011). While much of this literature has focused on the influence of father absence on daughters’ reproductive maturity, some studies that have looked at other parent–offspring dyads have shown similar results (Sheppard & Sear, 2012; Bogaert, 2005; Sheppard, Garcia & Sear, 2014). These patterns seem to contradict many evolutionary anthropological accounts that emphasize the importance of downward intergenerational investments (Kaplan, 1996), including from fathers (Hill, 1993; Geary, 2000) and from grandmothers in helping raise dependent offspring (Hawkes, 1998) since one way kin help can improve one’s fitness is by expediting reproduction.

In this paper we suggest that models of intergenerational conflict within cooperatively breeding kin groups can help explain why parental absences often expedite an adolescent’s reproduction. While intergenerational conflict can stem from several kinds of discrepancies between what parents and offspring want, we focus on parent–offspring conflict over reproductive opportunities. Humans’ ability to cooperate in raising altricial and slow-developing young that are born in relatively short succession has been proposed as part of the explanation for their success as a species (Hrdy, 2009; Bell, Hinde & Newson, 2014). This requires that some individuals invest in raising the offspring of other individuals, at least at some points during their life course. We would expect, therefore, that at least some individuals within the cooperative unit pay a short-term fitness cost, but that the cooperative unit in general benefits long-term from this exchange of resources or help. It should be noted that conflicts over public goods often occur within cooperative systems, meaning that cooperation and conflict are not opposite strategies, as their common usage implies. While “intergenerational negotiation” might be a better term for this phenomena, we will stick to the commonly used terminology in the literature of “intergenerational conflict”. In many species of cooperative breeders older siblings help care for young, thus delaying their own dispersal and possibly paying short-term costs in terms of their own reproduction (Jennions & Macdonald, 1994). A similar intergenerational cooperative arrangement has been proposed as a feature of many human societies given the extent of allocare that older siblings provide (Kramer, 2005; Crognier & Baali, 2001).

Biologists have developed several models examining the allocation of reproductive opportunities within cooperative interactions (Vehrencamp, 1983; Reeve & Keller, 1995; Reeve, Emlen & Keller, 1998), including the circumstances under which intergenerational conflicts are resolved in favor of parents’ vs. offsprings’ reproduction (Johnstone & Cant, 2010). If parents win such reproductive conflicts, we would expect that offsprings’ reproduction will be delayed if the older generation uses up alloparental and household resources that the younger generation would also need in order to reproduce. If the younger generation wins intergenerational reproductive negotiations over who breeds, we may instead see that parents forgo reproduction allowing their offspring to commence their own reproductive careers. While biologists working with cooperatively breeding species have attempted to apply this logic to human family systems (Emlen, 1995), these insights have been neglected in the parental absence and human life history literature (see Surbey, 1998 for a notable exception).

Common explanations of parental absence effects

Parents as cues

The most popular explanations of why parental presences delay first births focus on the possibility that these serve as cues to socio-ecological parameters. We will call these the “parents as cues” accounts. One such explanation suggests that parental absences indicate high extrinsic mortality risks in an environment, meaning that a developing child should reproduce sooner to reduce their risk of dying childless (Chisholm, 1993). In such environments, delaying reproduction in favor of growth, development and skill acquisition may not yield sufficient long-term fitness benefits because of mortality risks that the individual cannot easily control (Stearns, 1976). An alternative proposal is that parental—especially paternal—absences may indicate that low investment in parenting, low partner selectivity, and earlier reproduction are adaptive mating strategies in one’s environment (Draper & Harpending, 1982; Ellis, 2004). A problem with this latter explanation is that it is ambiguous why a scarcity of highly investing partners should necessarily lead to earlier ages at reproduction. A more general criticism of these proposals that parents are cues to environmental circumstances is that they seldom make clear why parents specifically, as opposed to a developing child’s broader social network, should be privileged as informative about locally adaptive life history and mating strategies in their adulthood (Nettle, Frankenhuis & Rickard, 2012). In fact, whether children learn much from their parents rather than from peers and non-kin is questionable (Harris, 1998).

Another version of the “parents as cues” explanation now treats parental absence as just one of many stressors which result in rapid reproductive maturation. Psychosocial acceleration accounts of life history development, first proposed to help explain why father absences should expedite reproduction (Belsky, Steinberg & Draper, 1991), have been broadened to include any stressors that might serve as indicators that one will live in a harsh environment and therefore should reproduce sooner (Belsky, 2012). All of the “parents as cues” models have been critiqued given their assumptions that environments are variable enough to merit developmental plasticity, but stable enough for childhood environments to be predictive of adult ones. The extent to which early childhood environments are predictive of future ones is an area of active debate (Wells, 2007; Rickard, Frankenhuis & Nettle, 2014).

Parent–offspring interactions

Two other kinds of explanations focus more directly on how parent–offspring interactions, rather than parental absence as a cue to socio-ecology, should influence reproductive timing. The first set of these “parent–offspring interaction” models focuses on parental investments. Ellis (2004) has proposed that children growing up in households with high quality care stand to benefit from capitalizing on this care by investing in their own growth (including skills), rather than in early reproduction. This means that children with parents present in their households would experience later ages of reproductive maturity, insofar as parental presence is a proxy for quality of care. A related argument suggests that fathers invest in, and guard, their daughters in ways that help them obtain high status and stable mates at the expense of earlier reproduction (Flinn, 1988). These arguments seem functionally plausible, especially given the importance of extended childhoods and slow life histories in humans (Kaplan et al., 2000), suggesting potential long-term fitness benefits to delaying reproduction. However, these “parent–offspring interaction” models focusing on the effects of parental investment deemphasize the possibility of parent–offspring conflict regarding help allocations, and assume the importance of downward, rather than upward, intergenerational transfers.

The second kind of “parent–offspring interaction” hypothesis focuses on inbreeding avoidance. This one suggests that reaching sexual maturity in the presence of parents poses a risk of inbreeding depression (Matchock & Susman, 2006). While this model has been useful for predicting reproductive strategies in cooperatively breeding species with small kin groups and high reproductive skew (Cooney & Bennett, 2000), we are more skeptical that inbreeding avoidance was an important selection pressure favoring delayed maturity in recent human evolutionary history given that adolescents can find unrelated members of the opposite sex with whom to reproduce even in small human social groups. Furthermore, in other primates with similar multi-male multi-female groups, individuals manage to largely avoid parent–offspring mating despite long alpha male tenures (Muniz et al., 2006).

An extension of the first “parents–offspring interaction” account suggests that parental deaths or separations may have consequences for life history development, but only insofar as they belong to a broader set of stressor that affect a child’s health outcomes. Several kinds of social strains during childhood—e.g., residential moves (Nettle, Coall & Dickins, 2011; Clutterbuck, Adams & Nettle, 2014) and chronic illnesses (Waynforth, 2012)—expedite maturation and reproduction. These kinds of stressors may change a developing child’s physiological status in such a way that the child becomes increasingly susceptible to age-specific extrinsic mortality and morbidity. This was originally proposed as the “weathering hypothesis” (Geronimus, Bound & Waidmann, 1999). Individuals whose health deteriorates rapidly with age need to get on with reproduction relatively early, in order to ensure successful childbearing while still relatively healthy. A key assumption in this argument is that developmental insults—including those resulting from lower parental investments—change an individual’s physiological state in a way that the individual himself would not be able to repair, or in a way that is not worth repairing (Rickard, Frankenhuis & Nettle, 2014). If this were not the case it is unclear why an unhealthy individual would not instead try to improve his physiological state at the expense of earlier reproduction. Additionally, the effects of father absences on age at first birth have been found to be greater than those of mother absences (Sheppard, Garcia & Sear, 2014). This result is counter to the more straightforward prediction from a “weathering” model given that mothers’ deaths tend to have more deleterious consequences for children than fathers’ deaths (Sear & Mace, 2008).

A confounded relationship

It is worth considering that parental absences may not lie on any causal pathway affecting reproductive timing. Rather parental absence effects may be confounded by other intergenerationally correlated genetic or social factors that affect both parental availability and an individual’s mortality risks, especially in post-industrial societies with steep socio-economic gradients in health outcomes . However, several studies have found robust parental absence effects on reproductive acceleration when controlling for socio-economic confounds (Ellis et al., 2003; Michael & Tuma, 1985; Kiernan, 1992; Vikat et al., 2002; Sheppard, Snopkowski & Sear, 2014), when controlling for family-level effects (Ermisch, Francesconi & Pevalin, 2004; Tither & Ellis, 2008), when examining internationally adopted children raised in well-off families (Teilmann et al., 2006), or when taking advantage of natural experiments during which parental absences have more extrinsic causes such as war or natural disaster (Pesonen et al., 2008; Cas et al., 2014). All of these studies suggest a unique causal role of parental absence above and beyond the confounding effects of genetic or socio-economic variables.

Here we resurrect the idea that intergenerational conflict may help explain parental influences on the timing of reproduction (Emlen, 1995), and suggest that models of intergenerational conflict complement, and have several advantages over the more common accounts of parental absence effects outlined above. First, unlike the “parents as cues” models, they can help account for the primacy of parents’ presence in explaining children’s reproductive timing. Second, intergenerational conflict models integrate the importance of conflict within cooperative systems into the extant “parent–offspring interaction” models that emphasize downwards parental investments, skill acquisition, and delayed maturation as important phenomena throughout human evolutionary history. Third, intergenerational conflict models allow us to make additional predictions about how parental effects on reproduction should vary cross-culturally within humans.

Models of intergenerational conflict

Other evolutionary scientists have fruitfully used intergenerational conflict models to illuminate human family dynamics. For example, tug-of-war models, where actors expend resources to compete over reproductive opportunities have been developed to explain the evolution of menopause (Cant & Johnstone, 2008; Johnstone & Cant, 2010) and the higher rates of intergenerational male conflicts observed in polygynous societies (Ji, Xu & Mace, 2014). These particular versions have treated genetic relatedness within populations as extrinsic features of the environment resulting from sex-specific dispersal patterns or varying degrees of local reproduction. They have also assumed that one sex controls reproductive decisions. For instance, in the model of human menopause evolution (Cant & Johnstone, 2008), in order for reproductive cessation to be favored among older women, mothers-in-law and daughters-in-law must compete with each other and resolve this conflict over limited household resources, meaning that women are controlling reproductive decision-making in a setting with substantial female-biased dispersal. In such a context daughters-in-law win the conflict because they are less genetically related to group members than the mother-in-law is, and thus suffer greater inclusive fitness opportunity costs to not reproducing. Another recent model of intergenerational and sibling conflict over parental resources suggests that parents may be more selective than their daughters over the latter’s mate choice if parents have to compensate for non-investing sons-in-laws (van den Berg et al., 2013). This dynamic implies that parents should pressure their children to be more selective, and therefore possibly slower, to choose mates than they would otherwise be, although the authors do not make predictions about parental effects on reproductive timing per se. Furthermore, this model does not allow upward intergenerational transfers and does not examine the tradeoff between a parent’s own and their offspring’s reproduction. Some other researchers have suggested the importance of intergenerational conflict in negotiating young adults’ reproductive strategies but have not modeled their hypotheses formally (Hoier, 2003; Surbey, 1998; Waynforth, 2002; Apostolou, 2012). We therefore contribute a formal, but simple, model of intergenerational conflict, and use it to predict the effect of family structure on reproductive timing, and the cross-cultural variation in parental effects.

Integrating models of intergenerational conflict and parental absences

We describe a general framework for exploring reproductive timing decisions within individuals’ lifetime, which makes no assumption about dispersal patterns or about the sex that controls reproductive decisions. More specifically, we examine under what circumstances a parent should win potential intergenerational reproductive conflicts, and have another infant, and under what circumstances their adolescent child should win the conflict and start their reproductive career. This setup parallels that used in the animal dispersal literature where offspring have to choose whether to leave their natal nest or territory (Koenig et al., 1992). We model the effects of (1) parental continuity (i.e., the probability that a parent does not switch mates), (2) costs to reproductive overlap, and (3) reproductive senescence (i.e., aging that results in the older generation producing lower quality infants compared to the younger generation). We investigate these parameters because of their relevance to the human literature on life history and parental presence, although they may speak to similar effects in, and across, other species. Mate-switching plays a large role in explanations of father absence effects on reproductive timing in humans (Draper & Harpending, 1982; Shenk et al., 2013), and varies significantly cross-culturally with mating system. Additionally, reproductive senescence is a topic of much interest for evolutionary anthropologists given women’s long post-menopausal lifespans (Hawkes & Coxworth, 2013), and the possibility of comparable reproductive cessation for monogamous men. Reproductive senescence for pre-menopausal women (Fretts et al., 1995) and men (Plas, 2000) has also been shown to affect infant survivorship and health outcomes. In this model we treat reproductive senescence as an extrinsic parameter that constrains reproductive decision-making. In other words we do not allow senescence to evolve, although other models have examined the extent to which sex-specific age and genetic structure in a population can select for reproductive senescence (Johnstone & Cant, 2010).

Factors affecting mate stability, costs of intergenerational reproductive overlap, and reproductive senescence of a parent relative to an offspring are likely to vary both within, and between, human populations, making this framework particularly useful for making predictions about how parental effects on children’s life history should vary cross-culturally.

We return to how our model of intergenerational conflict can contribute to our understanding of why various forms of parental absence in childhood may expedite reproductive maturity in humans in the discussion. There, we also develop several predictions regarding how cultural institutions may moderate these effects across human societies. However, first we describe the formal framework. In the next section we describe the setup for a simple game theoretic model including the payoffs to parents and their children of reproducing or not, given the other actor’s reproductive behavior. We then analyze the implications of the model in two stages. First, we model what each actor would do given that the other has reproduced. Second, we use these results from the first stage to model how much each actor loses from not reproducing first. Using these results we can determine under which circumstances parents or their adolescent offspring are likely to win intergenerational conflicts and reproduce.

A Simple Model of Intergenerational Conflict

Actors

In this model we assume there are two actors of reproductive age; a parent and her/his adolescent offspring who has yet to reproduce. We do not explicitly model mate search costs, instead assuming that the younger generation can acquire a reproductive partner should she want to. However, for simplicity we assume that we do not have to consider the strategic interests of the younger individual’s potential partner. Not only does this simplification keep the model tractable, we also believe that the decision to seek reproductive opportunities and mates should be modeled in its own right since an adolescent can invest in physiological and behavioral strategies that facilitate reproduction before marrying. This model is therefore analogous to models of dispersal decisions in non-human animals. We also assume that the parents’ other children do not affect the payoff structures below. We will refer to the older generation as the parent, or G1, and the younger generation as the adolescent, or G2. The sex of the actors does not qualitatively change the results.

Setup

The parent and adolescent must each decide whether to reproduce at a given point in time. Their decisions can result in one or two infants in the household. The payoffs to each actor of reproducing will depend on the other’s decision given that reproduction and infant survivorship or quality are affected by access to resources that are shared within a household. If resources were not shared within a family or household unit, then the actors would be competing with all other group members when deciding whether to reproduce and therefore would have little incentive to curtail their reproductive efforts even if their kin specifically had reproduced. The limited household resources may include alloparenting or caloric production, for example.

We model independent sequential decision-making in stage 2 of the analysis rather than synchronous decision-making that is blind to the other’s behavior. This is because parents and adolescents are likely able to detect each other’s reproductive effort and adjust their decisions accordingly, and because they have some incentive to communicate their intentions to coordinate their reproduction in this game (Cant & Shen, 2006). We further assume that actors have equal competitive abilities such that those who are willing to expend more competitive effort are more likely to assure themselves the right to reproduce first. This suggests that those who stand to lose more from forgoing reproduction should be more willing to compete for the right to reproduce first.

Parameters

We model the effect of three parameters; infant survivorship when G1 and G2 both reproduce relative to when only one reproduces (s), the relative fitness of an infant born to the younger generation compared to the older parent (y), and parental continuity (c). We define s as the ratio of survivorship of an infant who shares a household with another infant, relative to his survivorship being the sole infant in the household. This can take values from 0 to 1, where 1 indicates equal survivorship whether or not the infant shares his household with another infant; values less than 1 indicate lower survival if the infant shares his household relative to being the only infant. We assume there are never benefits to infants sharing a household, because they are competing for the same scarce resources. The parameter y (youth benefit) is the ratio of the fitness of an infant born to the adolescent relative to the fitness of an infant born to the older parent. We include this parameter in the model to allow reproductive senescence that can switch the resolution of the intergenerational conflict to favoring the younger generation. While y can take any positive value, we will primarily focus on values of y ≥ 1, where 1 represents equal fitness for the offspring of the older and younger generation, and y > 1 represents higher fitness for the offspring of the younger generation relative to the offspring of the older generation. While reproducing when very young can have detrimental effects (Chen et al., 2008; Fraser, Brockert & Ward, 1995), any fitness costs to infants of young parents (i.e., where y < 1) favor the older generation’s reproduction further and thus will only exaggerate the resolution of the conflict in favor of the parent. Finally, parental continuity, c, is the probability that G1 continues to reproduce with the same person who produced G2. This parameter only affects the adolescent’s, G2’s, payoff function. This continuity value, c, can also take values from 0 to 1, where c = 1 denotes that G2 will have a full sibling, and c = 0 denotes that G2 will have a half sibling.

Payoffs

Each individual can choose to reproduce R, or not N. Variables subscripted 1 denote payoffs to the parent, G1, while those subscripted 2 denote payoffs to the adolescent, G2. We denote the payoffs to each actor, V, using conditional probability notation. In each equation the first term represents the contribution of the actor’s own reproduction to her fitness and the second term refers to the other person’s contribution the actor’s inclusive fitness. The payoffs for each individual—G1 and G2 in sequence—when both reproduce are: (1) V1R|R=0.5s+0.25sy

(2) V2R|R=0.5sy+0.25s1+c.

The payoffs for each individual when only the parent, G1, reproduces are: (3) V1R|N=0.5

(4) V2N|R=0.251+c

and when only the adolescent, G2, reproduces: (5) V1N|R=0.25y

(6) V2R|N=0.5y

and, just for completeness, when no one reproduces: V1(N|N) = V2(N|N) = 0.

Results

What would each actor want given that the other one has reproduced?

Obviously, everyone wishes to avoid a household where neither actor reproduces. However, it is not always the case that both generations reproducing simultaneously maximizes each individual’s inclusive fitness. Under these circumstances, the payoff structure described in Eqs. (1)–(6) suggests that, for some part of the parameter space at least, parents and adolescents are engaged in a hawk-dove game. That is, this decision-making requires coordination so that the household does not end up with too many or too few infants, but at least some of the time each actor prefers to be the one to reproduce. Here we address the question of how each individual would respond were the other actor to have reproduced. Below we also show whether each actor would want the other individual to reproduce given that they themselves had already reproduced.

Given that the parent, G1, has reproduced

Under these circumstances the adolescent will want to reproduce when V2(R|R) > V2(N|R). This is true when: (7) s>1+c/2y+1+c.

However, the parent will only want her offspring to reproduce when V1(R|R) > V1(R|N). This is true when: (8) s>2/y+2.

Given that the adolescent, G2, has reproduced

On the other hand given that the adolescent has reproduced the parent will want to reproduce when V1(R|R) > V1(N|R). This is true when: (9) s>y/y+2

whereas, the adolescent will want her parent to reproduce only when V2(R|R) > V2(R|N). This is true when : (10) s>2y/2y+1+c.

Summary of payoffs to adding a second infant to household

The first column of Fig. 1 shows the parameter space over which actors want the adolescent, G2, to reproduce given that the parent, G1, has done so, assuming a reproductive benefit to the older generation (y = 0.3), no youth benefit (y = 1), and a threefold youth benefit (y = 3). These are represented by the areas above the line for each actor. First focusing on Fig. 1C, when neither generation has a reproductive advantage (y = 1), if the survival ratio of 2 to 1 children in the household, s, is high enough there will be no conflict of interest as both actors will want the adolescent to reproduce. Similarly, if s is low enough neither actor will want the adolescent to reproduce because the additional infant will decrease the survival odds for both children too much. Disagreements between parent and adolescent in terms of adding a second infant to the household arise for intermediate values of s. The straight line for the parent shows that she has a higher threshold s for her to want her offspring to reproduce, and that this value does not depend on c since parental continuity does not affect a grandparent’s relatedness to her grandchild. The adolescent′s line on the other hand increases with c, that is the higher her relatedness to her new sibling the higher the survival ratio (s) has to be in order for her to benefit from reproducing as well.

Figure 1 When actors should want a second infant in the household given one of them has already reproduced.

Areas are plotted as a function of survival ratio (s), parental continuity (c), and youth benefit (y). Areas above each actors line denote when it is in their fitness interest to add the second infant to the family. The text within the plots denotes which actors want the second infant.

Figure 1D shows the same lines for each actor given that the adolescent has reproduced. Now the adolescent has a higher threshold of survival ratio for which she would want her parent to reproduce compared to the parent′s own threshold. As her certainty that she will get a full sibling (c) increases, the adolescent becomes more tolerant of her parent′s reproduction, that is, she benefits from a sibling for a wider range of costs to having two infants in the household (s). Still, even if the adolescent is a full sibling of the parent′s child, there will be values of s for which she will not want her parent to reproduce even though the parent wants to.

By increasing the youth benefit, y (Figs. 1E and 1F) the parent has relatively more to gain from a grandchild. This reduces the size of the zone of conflicts of interests in both scenarios, but maintains the order of the lines in Fig. 1. Both lines move down in the first column, and up in the second one. That is, both actors will want the adolescent to reproduce over a wider parameter space given that the parent has reproduced, whereas both actors will be more reticent to encourage the parent′s reproduction once the adolescent has already reproduced.

Similarly, when the youth benefit is less than 1 (Figs. 1A and 1B), the parent has a reproductive advantage and the zone of conflict over the second child becomes smaller. The threshold lines in the first column of Fig. 1 are higher when y = 0.3 meaning that both actors are less willing to have the adolescent reproduce given that she would produce a relatively lower quality infant. On the other hand if the adolescent has reproduced, both actors are more tolerant of the parent reproducing when the older generation’s reproduction is more efficient as shown by the threshold lines being lower in (B) relative to (D). Given that values of y < 1 further favor the parent’s reproduction we will not focus on this part of the parameter space.

It should be noted that even when both actors agree that a second child should not be added to the household, there may be conflict over whose child that should be. That is, for G1, V1(R|N) is better than V1(N|R) so long as y < 2. In other words, if only one person is going to reproduce the parent prefers to be the one to do so, as long as the youth benefit is less than 2. Similarly, the adolescent, G2, prefers to be the one to reproduce much of the time. For example, when there is no reproductive benefit to the older parent reproducing (y ≥ 1), V2(R|N) is strictly better than V2(N|R), so long as they are unsure that their parent will produce a full sibling—i.e., c < 1. Again, these hawk-dove dynamics suggest the importance of competition and coordination among the actors.

How much do actors lose from not reproducing first?

The first stage of analysis presented in the previous section shows that there are conflicts over who gets to reproduce, and that the actors will not always agree about adding a second infant to the household given that one of them is already giving birth. We now assess how much each actor stands to lose by not reproducing first. We assume that the actor who stands to gain more from reproducing first has more to gain from expending competitive effort to assure herself the first mover position, and thus her preferred reproductive outcome. It is worth noting that actors may exhibit fitness losses to reproducing first, meaning that they prefer to choose their strategy after the other actor has done so, and thus do not need to compete over the right to be the first mover.

We use payoffs from the previous section regarding what actors would do as second movers to calculate the payoffs to each actor were the parent, and subsequently were the adolescent, to reproduce first. We assume the second actor has full autonomy in their decision so that, even though we plotted what both actors wanted in Fig. 1, only the function for the second actor matters.

Payoffs to actors if the parent, G1, reproduces first

The adolescent, G2, will be the second actor and will respond differently to G1’s initial decision, depending on the values of s, y and c. Therefore, we need two different functions to determine the ultimate payoffs for each generation, depending on what the adolescent does. (11) V1=V1R|R=0.5s+0.25syif s>1+c/2y+1+c,V1R|N=0.5if s<1+c/2y+1+c.

(12) V2=V2R|R=0.5sy+0.25s1+cif s>1+c/2y+1+c,V2N|R=0.251+cif s<1+c/2y+1+c.

Payoffs to actors if the adolescent, G2, reproduces first:

The parent, G1, will act differently depending on whether s is greater or less than y/(y + 2). (13) V1=V1R|R=0.5s+0.25syif s>y/y+2,V1N|R=0.25yif s<y/y+2.

(14) V2=V2R|R=0.5sy+0.25s1+cif s>y/y+2,V2R|N=0.5yif s<y/y+2.

Summary of costs to not reproducing first

The parent, G1, will want to reproduce first when Eq. (11) > Eq. (13), and the adolescent will want her parent to reproduce first when Eq. (12) > Eq. (14). As a simple example, let’s consider payoffs when c = 0 and y = 1. In this case, both actors will always want to be the first mover, or at worst be indifferent if s > 1/3 since both of them will reproduce when there are low costs to both reproducing. When s < 1/3, each actor will lose 0.25 if she does not get her way. In other words the game is symmetric, and it is not obvious who will win the conflict. This is not surprising as when c = 0 both actors are equally related to the other actor’s child. In much of the parameter space, however, the game is not symmetric, and one actor stands to lose more than the other by not reproducing first. Here, we can identify the most likely winner of the conflict, namely the one who stands to gain more from being the first reproducer.

Figure 2 illustrates the fitness losses to each actor as a function of whether they get to reproduce first or choose their strategy after the second actor for a broader set of parameters. In a tug-of-war model, the fitness losses would correspond to how much actors should be willing to invest in competitive effort to win this conflict. This means that the higher an individual’s opportunity costs to not reproducing first relative to the other actor’s opportunity costs, the higher her likelihood of winning the conflict. The horizontal axes show that these conflicts will be resolved differently as a function of the costs to having two infants in the household, s (note: s values have changed from being on the vertical axis in Fig. 1). Each plot represents a different combination of youth benefit, y, and parental continuity, c.

Figure 2 Cost of choosing strategy after the other actor has reproduced.

Fitness losses are plotted as a function of the survival ratio, s, of 2 vs. 1 infant in the household. The solid line denotes the older generation (i.e., the parent), and the dotted line denotes the younger generation (i.e., the adolescent). The larger the fitness losses from not reproducing first, the more likely the actor is to win the conflict. Values of c = 0.25 and c = 1 represent low and high parental continuity respectively, and increasing values of y represent higher fitness of the younger generation’s infant. When y = 1 there is no senescence. Intermediate y values correspond to values of y=1+c for c = 0.25 and c = 1 respectively. At these values the payoffs work out such that the actors never disagree about whether there should be two or one infant in the household. Plot (H) shows the limits of three functionally different zones; zone (1) where only one actor will reproduce, (2) where the number of infants produced will depend on who reproduces first, and (3) where both actors will reproduce. The corresponding zones can be found in all other plots except for (C) and (F) where zone 2 disappears.

There are three areas of the parameter space that have functionally different outcomes for the set of actors (illustrated in Fig. 2H). We will discuss these out of order from simplest to most complicated (starting with zone 3, ending with 2). Zone 3 corresponds to survival ratios, s, that are high enough such that both actors will reproduce regardless of who acts first. This means that both actors have zero fitness losses to choosing second in this zone, since everyone will get to reproduce. This corresponds to the right hand side of each plot in Fig. 2. On the left hand side of each plot is zone 1 where the survivorship of two infants relative to one, s, is so low that only one actor will reproduce. Finally, in zone 2 with intermediate survival ratios, the number of people who reproduce will depend on who reproduces first. This area is indicated by the sloping lines in Fig. 2 and requires more explanation. This intermediate area is bounded by s = (1 + c)/(2y + 1 + c) and s = y/(y + 2), as outlined in Eq. (11) through (14). Whether each of these expressions denotes the upper or lower limit of the intermediate area depends on the values of y and c. For example, when there is no youth benefit, y = 1, (1 + c)/(2y + 1 + c) > y/(y + 2). This means that the younger generation has the higher threshold s value at which they would produce a second child, and is therefore more reticent to reproduce given that the other actor has already done so. However, this will flip when the younger generation’s reproduction is significantly more efficient—i.e., for large enough values of y, specifically when y>1+c. Under these circumstances the parent will have a higher threshold value for reproducing as a second mover than the adolescent does. When y=1+c zone 2 disappears (e.g., Figs. 2C and 2F), meaning that parents will not have a different preferences for total infants from adolescents as second movers.

We first elaborate on the conflict dynamics using the simple case where there is no efficiency benefit to the younger generation reproducing, y = 1. In this case, when the costs to synchronous reproduction is low enough (e.g., s > 1/2 in Fig. 2B) both individuals will end up reproducing meaning that order of decision-making is irrelevant. The lower c is, the larger this parameter space, as indicated by the longer range of zero fitness losses (zone 3) of Fig. 2A than 2B. This means that with lower probabilities of parental continuity, the greater the range of survival ratios under which the adolescent is willing to reproduce. If the costs of synchronous reproduction are high enough (s is low), only the first actor will reproduce and the parent stands to lose more than the adolescent from not being the one to do so (see zone 1 of Figs. 2A and 2B). In fact if c = 1 the younger generation should be indifferent between reproducing or having their parent produce a full sibling. This is indicated by the zero fitness loss to the adolescent of choosing not to reproduce after the other individual has. For intermediate values of the survival ratio, s, the actors pursue different strategies as 2nd movers. In the case of y = 1, in this intermediate range the parent will prefer to reproduce whether or not the adolescent has done so, whereas the adolescent would want to reproduce only if the parent does not. This explains the negative “losses” to going 2nd for the adolescent, who prefers to decide not to reproduce after having seen the parent reproduce, than to reproduce herself first and then have the parent add a 2nd child to the household. Under these circumstances (and whenever one actor suffers negative losses from going second in Fig. 2) the model exhibits endogenous timing, meaning that both players agree about who should act first (Cant & Shen, 2006). Given the costs to pregnancy termination and the mutual interests of kin, such contexts should favor signaling reproductive intent.

Once we add large enough reproductive consequences to senescence (e.g., Figs. 2G and 2H where y = 3), the younger generation wins out over the parent during contexts of reproductive conflict. In the intermediate zone 2, if senescence is high enough (y>1+c) both actors prefer the younger generation to reproduce alone, given that the adolescent would reproduce regardless of the parent′s reproductive decision in this range. For really severe resource constraints (zone 1) the bottom two rows of Fig. 2 show that the younger generation will also lose more from going second and not reproducing than the parent will. In fact, for very large youth benefits (e.g., y = 3) even the parent prefers the adolescent to be the sole reproducer as indicated by her fitness losses to going second being negative. This indicates that for this part of the parameter space, even as first mover, the parent would forgo reproducing in favor of allowing her child to do so. More generally this should be true when y > 2. However, in zone 1, the range of y values for which the adolescent stands to lose more than parent is even broader. So long as y > (3 + c)/3, the parent loses less than her child from forgoing reproduction when only one of them is going to reproduce.

Generally, the model shows that the higher the probability of parental continuity c, the easier it is for the parent to win the intergenerational conflict, while it is more likely that the younger generation wins the conflict as y increases. For parts of the parameter space (i.e., when the lines fall below zero) it is even to an actor’s advantage to allow the other individual to reproduce first and forgo reproducing themselves. For intermediate values of the survival ratio, s, this is because the “losing” actor (e.g., the parent in Figs. 2E–2H) would not reproduce were there an infant in the household already, whereas the other actor would reproduce regardless. For low enough s both actors agree that only one individual should reproduce, which creates the discontinuities in fitness loss values.

Discussion

Our model suggests that parents and their children will often agree about reproductive decisions when there are low costs to synchronous reproduction, but that parents will generally have the upper hand in negotiating intergenerational conflicts should these arise (i.e., when s is low enough). However, this is contingent on the adolescent’s expectation of her parent producing a full-sibling. This means that biparental presence should favor the parents’ reproduction over their offsprings’ reproduction and may thus delay the latter’s age at first birth. This dynamic is driven by the asymmetric relatedness of actors to the potential offspring being produced. However, this game becomes fully symmetric if the parent cannot give birth to a full sibling, meaning that if one parent is absent, offspring should be as likely as the remaining parent to win intergenerational reproductive conflicts. It follows that relative to having two parents present, an adolescent has more of an incentive to reproduce when one parent is absent since her future siblings will be less related to her. Furthermore, the advantage can even shift to the younger generation’s reproduction if we incorporate physiological senescence that reduces the quality of the older generation’s child.

It is worth noting that these predictions hold only for species where parents and offspring cooperate for reproductive purposes or rely on the same resources to reproduce. Similar facultative helping-at-the-nest as a function of relatedness to siblings has been documented among cooperatively breeding birds (Komdeur, 1994), suggesting these helpers also disperse and reproduce at a later age. Furthermore, experiments with eusocial Damaraland mole rats show that switching out a related dominant male from the family group induces physiological changes and reproductive activity among the dominant pairs’ daughters (Cooney & Bennett, 2000). Although we do not argue that humans are similarly eusocial, this line of evidence suggests that analogous physiological and behavioral pathways may help explain observed changes in human menarche (Webster et al., 2014), adrenarche (but see Sheppard & Sear (2012) showing father absences in late childhood may delay male puberty), and age of first reproduction when parents are absent.

In the remainder of the discussion we develop predictions both about (1) factors that affect the likelihood that offspring win intergenerational reproductive conflicts and therefore start reproducing, and (2) factors that affect the size of the parental absence effect on offsprings’ reproduction. Figure 3 illustrates the difference between these two kinds of predictions. We return to this distinction and explain it further later. First, we consider how our model speaks to the effect of different kinds of parental absences on reproductive maturation. Then we turn to the effects of sex, gender, age and other cultural institutions on intergenerational relations.

Figure 3 Effect of other parent’s absence (parental continuity, c, is 0 rather than 1) on adolescent’s willingness to reproduce if her parent has done so, as a function of youth benefit.

Green and red curves denote minimum survival ratios (s) at which adolescent will reproduce, derived from Eq. (7). When the second parent is absent, the adolescent will reproduce for a larger part of the parameter space. The dotted grey line (plotted on the right axis) shows that the relative size of the parental absence effect increases with the youth benefit (y)—i.e., the difference between these threshold levels of s when c = 1 relative to c = 0 as a percent of the effect when c = 1 increases with y. Vertical lines mark the y-values above which the parent becomes more reticent to add a 2nd infant to the household than the adolescent for the respective parental continuity values.

Predicted differences in parental absence effects by type of absence

Our model does not explicitly differentiate households with a stepparent from those with a single parent. This is because we assumed that both generations could find a mate at no cost and that other actors’ interests did not matter. Table 1 shows how different kinds of family compositions correspond to three parameters; parental continuity (c), the remaining parent’s mate search costs, and the presence of a non-kin actor in the household. The first column shows that if we only consider parental continuity we should expect about equal effects of parental absence due to divorce when there is a stepparent present as when the absence is due to death. In contrast, a child raised in a single parent (non-widow) household should not expedite their reproduction as much as children raised in other parent absent households given there is some chance of parental continuity (c > 0). The child may rely on other cues to parental continuity under such circumstances, such as degree to which the absent parent invests or visits, to assess the probability of parental continuity. This may explain the fact that father absences due to labor migration do not expedite adolescent’s reproduction (Shenk et al., 2013), that tense mother–father relations (Chisholm et al., 2005) and residential moves (Clutterbuck, Adams & Nettle, 2014) expedite maturation, and that the quality of paternal care matters more to pubertal timing than mere presence (Ellis et al., 1999).

Table 1 Effects of different kinds of parental absences.

Effect of family structures on parental continuity (c), the presence of a non-kin actor in the household, a parent’s experiencing mate search costs, and an adolescent’s predicted age at first birth (AFB). Note: When incorporating mate search costs, adolescents would always experience these as well.

Family household structure	Parental
continuity
(c)	Non-kin actor
present	Mate search
costs to
parent	Adolescent’s
predicted AFB	
Two genetic parents	∼1	No	No	Latest	
Parent and step-parent	∼0	Yes	No	Intermediate	
Widowed single parent	0	No	Yes	Earliest	
Other single parent
(e.g., labor migrant or separated)	>0	No	Maybe	Intermediate	

We can relax assumptions about mate search costs and other actor’s interests in our model to derive more predictions about family structure. If we incorporate mate search cost it is easy to see that a parent and adolescent are in the most symmetric situation when neither has a partner since both will have to pay the costs of finding a mate (see fourth column of Table 1) and both will produce infants that are 0.25 related to the other actor. While this symmetric relatedness to infants is the same in a stepparent present household, asymmetries that favor the parent arise when we incorporate mate search costs and the stepparent’s interests. Relative to being a single parent, a stepparent’s presence more clearly indicates a parent’s intention to reproduce. Not only does the presence of a stepparent mean that only the younger generation has to pay mate search costs, but it may also commit a parent to reproduce given that a stepparent has no inclusive fitness interests in his stepchild’s reproduction and therefore stands to lose a lot from not reproducing himself. This leads to the prediction that parental absences due to death should expedite an adolescent’s reproduction the most since this gives the younger generation the most leverage in family-level negotiations, especially in societies with large costs to marriage.

Previous accounts of how family structure affects intergenerational conflict over reproduction and age at first birth have relied on verbal arguments and therefore made ambiguous predictions that do not necessarily match those we have proposed above (Surbey, 1998; Apostolou, 2012). For example, Hoier (2003) suggests that “the maternal reproductive interests model also predicts an earlier menarche if the mother has children with the stepfather, but a less pronounced one because half-siblings are not as closely related” (p. 214). The family structures being compared are unclear in this formulation, but the author’s expectation regarding the effect of sibling relatedness on maturation seems contrary to our own. While her phrasing suggests that the lower relatedness among half-siblings would make parental absence effects smaller (i.e., expedite maturation less), we predict it is exactly because of this lower relatedness to half-siblings that stepfather presence should expedite adolescents’ reproduction. In another example, Apostolou’s (2012) verbal formulation suggests that parents should want their children to reproduce earlier than the children themselves would want to reproduce so that they can control the younger generation’s reproductive decisions more effectively while the children are younger and more dependent on parents. In this account it is unclear why it is not in the fitness interest of adolescents to reproduce as early as they are reproductively mature as well, and how the older generation’s political gains from controlling their children’s reproduction trades off with their gains from using their children’s labor and having healthy grandchildren.

Predicted differences in parental absence effects by actors’ sex and gender

While our model can apply to actors of either generation that are any sex, there are multiple reasons we might expect sons and daughters to be differentially affected by intergenerational conflict. We can conceptualize these sex and gender differences as changing the youth benefit. First, non-zero rate of paternity uncertainty will negatively affect a son’s fitness through both his own reproduction and through that of his siblings; whereas it will only affect a daughter’s fitness through her siblings. This means that sons should favor their own reproduction less than daughters would, and thus that they are less likely to win intergenerational conflicts over reproduction. This would be mathematically equivalent to sons having a lower youth benefit, y, than daughters. On the other hand, given that men tend to marry later than women do, their parents will be on average older when they commence reproductive negotiations, and thus more likely to lose reproductive conflicts given the larger physiological youth benefit, y, at these ages. This means that sons should be less likely to win intergenerational reproductive conflicts than daughters when there is high paternity uncertainty, and more likely to win intergenerational conflicts when men marry late.

Additionally, while we haven’t allowed parental coercion in our model, were we to do so it is possible that parents would be more likely to delay daughters’ or sons’ reproduction, depending on the relative contributions of each gender to the household. For example, if alloparental care is a scarce resource then parents might delay daughters more given that they more commonly help rear younger siblings cross-culturally, whereas if meat is a scarce resource, parents may delay sons more given that men are usually more responsible for procuring animal protein.

So far we have focused on how actors’ sex and gender roles can affect their likelihood of winning intergenerational reproductive conflicts, but have said nothing regarding how sex changes the size of parental absence effects. We can illustrate this interaction by considering more generally how changes to the youth benefit (in this case, those resulting from sex differences) alter the difference in the survival ratio at which an adolescent would be willing to reproduce when both genetic parents are present relative to when one of them is absent. Although larger youth benefits make it more likely that the younger generation wins intergenerational conflicts, this does not mean that it translates into a smaller parent absence effect. Figure 3 shows the threshold survival ratio value above which the adolescent will reproduce given that her parent has already done so, in a parent absent vs. both parents present household (c = 0 vs. c = 1). Both of these downward sloping curves show that the adolescent is willing to reproduce over a wider range of survival costs, s, to having two infants in the household as the youth benefit, y, increases. However, the dotted grey line shows that with larger youth benefits, the size of parental absence effects increases. Specifically, the percent difference in the threshold value of s for parent absent vs. parent present households increases with y. Substituting in the relevant sex differences for y, this implies that parental absence effects should be smaller for sons than daughters when there is high paternity uncertainty (ysons < ydaughters), but larger for sons than daughters when ages at first birth are much later for men (ysons > ydaughters).

It is worth noting that this model would also make similar predictions for mother and father absences under conditions of little paternity uncertainty. If we incorporate paternity uncertainty, the father’s reproduction is of relatively less value to an adolescent than that of her mother. This may help explain why helpers-at-the nest more often help mothers than fathers. It follows that the younger generation is more likely to win intergenerational negotiations with a single father than a single mother, and that mother absence effects should be larger than those of father absence. The emphasis in the life history literature on the effects of father absence may reflect the higher variance in paternal than maternal availability, both due to death and divorce. However, contrary to our predictions, or to the predictions we would derive from models focusing on parental investments, some data suggest that father absences have more expediting effects on reproduction than mother absences (Sheppard, Garcia & Sear, 2014).

Predicted differences in parental absence effects by actors’ ages

While we emphasized youth benefits greater than 1—i.e., physiological senescence such that only the parent’s infant could ever be lower quality than the offspring’s infant—any process that makes one generation’s infant higher quality than the other moves the resolution of the conflict in favor of that individual. Clearly, the older a parent is, the more likely the younger generation is to win this conflict, assuming that relatively elderly parents produce lower quality children (Fretts et al., 1995; Plas, 2000). Humans experience a particularly unusual pattern of senescence given that women’s reproductive system declines, while they are still healthy and productive adults, thus allowing them to shift strategies to alloparenting even if they lose intergenerational conflicts. However, in the other direction, the more benefits to learning parenting skills or to delaying development for an adolescent, the more likely the older generation is to reproduce, and the smaller the scope for intergenerational conflict. This means that the more an adolescent gains from delaying reproduction (i.e., the farther below 1 y is), the less a parental absence should affect her developmental trajectory. We see this pattern in Fig. 3 where the percent difference between parent absence and parent presence approaches 0 as the youth benefit approaches 0. This dynamic might be particularly important in humans given the large repertoires of cultural traits (including parenting skills) that they need to learn to become competent adults.

Predicted cross-cultural differences in parental absence effects

Here we develop predictions about how ages at first birth and how parental absences on age at first birth may vary cross-culturally. Institutions can affect both how likely members of the younger generation are to win intergenerational negotiations (corresponding to mean ages at first birth), and the effect of parental absences on timing of first births (corresponding to the difference between parent present and parent absent households). The effect that parental investments have on age at first birth are likely to vary cross-culturally as well—for example, father absences are likely to have smaller effects in societies where paternal investments are not as important. However, here we focus on how the effect of parental absences on reproduction should change cross-culturally due to the reduced relatedness of future siblings.

1. At the population level, in societies with less turnover between partners and less paternity uncertainty we would expect greater alloparenting or provisioning of younger children by older siblings or helpers-at-the-nest. This suggests there may be a group-level negative association between repartnering rates and ages at first birth. This also means that the presence of a father should delay reproductive maturity less in societies where mere presence is not a good indicator of his producing full siblings in the future, as might be the case in societies with partible paternity institutions or high female sexual autonomy.

2. Polygamous contexts where future siblings are less likely to be full siblings should similarly discourage the younger generation from investing in their natal household. Again this suggests children being raised in polygamous households may experience earlier reproductive maturity and that the presence of both parents may be a bad indicator of future returns to investing in the natal household. By this logic we might expect smaller parental absence effects in such households. It is also worth noting that a polygynous man can extend his reproductive career by acquiring new wives. This puts him in direct competition with his children—especially his sons—over household resources for bridewealth and over mates. This should increase intergenerational conflict, but it is less clear who should win these conflicts. If a polygynous father has some chance of reproducing with his son’s mother again, the father should retain more leverage in intergenerational negotiations, and thus delay his son’s reproduction, though less so than a monogamous man within a polygynous society.

3. Societies with bridewealth or dowry increase the mate search costs for the younger generation. This means that there are relatively higher costs to the younger generation reproducing with a given quality mate when bridewealth or dowry are expected. This should result in delays to the average ages of first birth and smaller parent absent effects for the affected gender, all else equal (see effect of smaller y-values in Fig. 3).

4. Similarly, there is cross-cultural variation in the degree to which parental contributions are needed to marry, set up a household (e.g., higher setup costs in neolocal societies than in patri- or matri-local ones), or to become competent and skilled adult members of society. Under these circumstances there is less intergenerational conflict over timing of first birth given the benefits to adolescents of skill and material acquisition. We would expect later mean ages of first birth under these circumstances. Again, the effect of parental presences relative to absences should be smaller given the smaller youth benefit in the reproductive domain. Importantly, this prediction is based on the assumption that the parental absence only affects an adolescent’s relatedness to future siblings, but not her other socio-economic or health outcomes (i.e., education, resources or embodied capital).

5. Ambilocal post-marital residence patterns may afford families the option of moving adolescents to contexts with less resource stress, effectively increasing the infant surival ratio s, thus reducing intergenerational conflict and allowing earlier ages at first birth. Furthermore, the greater options afforded in societies with flexible residence norms may give the younger generation more leverage in intergenerational negotiations. This may be true for neolocal residence norms as well, assuming costs to setting up a household are low. Such flexible residence norms should reduce the extent to which parents delay their children’s reproduction since both generations are likely to reproduce regardless of parental continuity rates when there are low costs to simultaneous reproduction (see high s values in Fig. 2).

It is worth discussing whether the intergenerational conflict model applies to low fertility, post-industrial societies where most of the empirical research has been conducted. In these societies several cultural norms reduce the extent to which we would expect intergenerational reproductive conflict. First, reproductive overlaps between parents and offspring are relatively rare in societies with late ages at first birth and early reproductive cessation. Second, cultural norms limiting child labour and fertility reduce the usefulness of adolescents to their natal household’s economy. Relatedly, while perceptions of the costs associated with rearing a given child may be increasing with expectations of high educational investment (Mace, 2008), these same institutions effectively decrease the reproductive youth benefit. Third, perceptions of household resource stress in most large-scale societies do not necessarily indicate an inability to raise reproductively successful adult offspring given the relatively low rates of infant and child mortality across socio-economic strata in modern economies.

This means that even if intergenerational reproductive conflicts do not account for parental absence effects in post-industrial societies, they may well help explain the phenomena in other cultural contexts. For example, it may be that in post-industrial societies parental absence effects are better explained by socio-economic health differentials, whereas in pre-transition societies intergenerational conflicts play more of a role. Alternately, it is possible that several of the psychological mechanisms implied by this intergenerational conflict model systematically misfire, even in low fertility societies, and result in maladaptive outcomes. If this were the case we would posit that adolescents have an evolved expectation of reproductive conflict with parents that does not accurately reflect reality in post-industrial settings. This misfiring account would imply a relatively canalized, rather than plastic, set of psychological mechanisms. A cross-cultural comparative approach may help disentangle some of the proposals on the table.

Limitations of the current model

While we made several simplifying assumptions to keep the project tractable future work can develop other avenues of inquiry. For example, one might extend the two person game to include the motivations of other potential actors, such as spouses for the younger generation. If the younger individual is betrothed or partnered, their spouse will have no inclusive fitness incentives to help raise their siblings-in-law. Such affinal ties only exist once the younger generation has married, a state suggesting that the parental generation may have lost intergenerational reproductive negotiations. This might help explain why the literature shows that a woman’s in-laws expedite first births more often than a woman’s parents do (Sear, Moya & Mathews, 2014). That is, given that the older generation has lost this intergenerational conflict, and their child has married, they may stand to gain from facilitating the production of grandchildren. Negotiations between other older siblings might also be of importance when deciding how alloparental care is provisioned, as has been shown in other cooperatively breeding species (Pasinelli & Walters, 2002).

It is also worth noting that the economic structure to the game we modeled might not reflect real world contexts if there are efficiencies of scale to raising two children together rather than two children apart. The extent to which intergenerational overlaps in reproduction are costly is a question of much empirical debate that has yet to be resolved (Lahdenperä et al., 2012; Mace & Alvergne, 2012; Skjærvø & Røskaft, 2013). We have also assumed that senescence is an extrinsic process rather than one directly under selection. There is some evidence that female reproductive physiology might be thus constrained (Robson, van Schaik & Hawkes, 2006), but this is debated given the diversity of senescence rates both within (Thomas et al., 2001; Snopkowski, Moya & Sear, 2014) and between (Jones et al., 2014) species.

There are also several reasons to believe that this model might underestimate the upper hand that the older generation has when the game has hawk-dove dynamics. For one, we assume autonomous decisions, whereas cross-culturally parents tend to have some coercive power over their offspring. This coercion may arise from dynamics beyond the genetic asymmetries illustrated in our model. In any case, it is likely that either group-level adaptive or non-adaptive cultural institutions play a role in the evolution of such norms. Second, caring for children, especially in humans, takes some specialized skills and the younger generation may stand to gain from the learning opportunities afforded by taking care of a child under the supervision of an experienced parent with a higher vested interest in the wellbeing of the infant. In fact, first born children are often at higher risk of mortality, both because of younger mother’s physiological development and relative inexperience (Hobcraft, McDonald & Rutstein, 1985).

While we have discussed this model in terms of intergenerational conflict, it is worth remembering that these family dynamics are being played out in a larger population of less related households. Bordered tug-of-war models incorporate pressures from between-group competition. These limit the extent to which group mmebers (e.g., kin) engage in costly internal conflicts (Reeve & Shen, 2006). Such models remind us that conflicts within cooperative units occur within a larger population of competitors, meaning that selection should favor reduced negotiation costs, and more efficient cooperative equilibria between parents and offspring. Cyrus & Lee (2013) have proposed that the division of labor regarding alloparenting and calorie production between the generations of human cooperative breeders is one such efficient equilibria that can be modeled as a multi-stage evolutionary process.

Conclusion

The model proposed here provides an explanation for why family structure specifically can result in different maturational rates and ages of first reproduction. We have argued that the intergenerational conflict model is more plausible than the popular “parents as cues” models in the literature, and a more complete account that complements the available “parent–offspring interaction” models. To summarize, parental absences in childhood and adolescence may provide cues of reduced inclusive fitness value to investing in future half-siblings rather than reproducing on one’s own. In contrast, if an adolescent perceives that her parents’ relationship is stable, she should be indifferent between reproducing on her own or helping rear any resulting full siblings. This could shape a developing child’s life history strategy, both physiologically (e.g., earlier menarche when a parent is absent) and behaviorally (e.g., earlier mate seeking and reproduction). This also provides a simple framework for devising predictions about how cultural and socio-ecological parameters should interact with family structure in affecting adolescents’ reproductive decisions.

Members of the Evolutionary Demography Lab, David Lawson and Kristin Snopkowski provided helpful comments on various versions of this manuscript. Daniel Nettle and Michael Cant offered critical useful suggestions as reviewers.

Additional Information and Declarations

Competing Interests

Author Contributions

The authors declare there are no competing interests.

Cristina Moya conceived and designed the experiments, performed the experiments, analyzed the data, wrote the paper, prepared figures and/or tables, reviewed drafts of the paper.

Rebecca Sear wrote the paper, reviewed drafts of the paper.

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
