# Peer review of "Intergenerational conflicts may help explain parental absence effects on reproductive timing: a model of age at first birth in humans"

_PeerJ, doi:10.7717/peerj.512_

## Round 0.1 · original submission · Minor Revisions

Please review the comments provided by two reviewers. Your manuscript was reviewed favorably, but there are some areas that need minor revision. Specifically, it would be useful for you to adequately engage the literature and to clarify the model. Both reviewers indicated that the model might be better situated contextually, with the first reviewer noting the introduction did not provide enough information to explain why your model is needed and the other reviewer noting a need to connect the abstract theory to biological meaning.

·

Basic reporting

The model is well explained and well reported (if at times the presentation - especially discussion and introduction - is a little on the verbose side for my taste). I have some substantive comments on the literature review (section 1), but I will bring those up under General Comments below. They do not affect the model itself.

Experimental design

N/A as this is theoretical paper, but the model seems appropriate and well explained.

Validity of the findings

Again, this is a theoretical paper, but I think it makes a very interesting argument I for one had not thought about before, and so it deserves some consideration at least as a possibility (see my General Comments below).

Additional comments

Major comments

My most major comment is that I like the model and find it interesting, but your current introduction does not persuade me that there is actually a phenomenon in need of an explanation here. That is, you would need to show that parental death/absence has a specific accelerating effect on reproduction, and is not just one more marker of general stress and disruption. Lots of other stressful things are known to accelerate reproduction in low-fertility societies, such as residential instability (which you mention), childhood illness (Waynforth 2012), poverty etc. These are not within the scope of your model. A plausible interpretation of these is just that early stress leaves you in physically worse shape (through the physiological wear and tear it imposes), meaning that you will weather less well with age and so have less time in which to complete your life-history goals. This makes you want to get on with it sooner. Arlene Geronimus made this argument a long time ago and there is a lot to be said for it. The parent absence correlations could fall within this too without requiring special explanation. There is lots of animal evidence too of early disruption leading to acceleration, for similar reasons (e.g. in kestrels, Cartwright et al. 2014 Current Biology).

I find the current page 2 a bit unsatisfactory as it notes the problems with the 'parents as cues' models, but does not mention the weathering approach (e.g. as developed by Rickard, Frankenhuis and Nettle 2014), which does not rely on the same assumptions as the 'parents as cues' approach. More particularly, page 2 mentions the idea that parental absence might be just one amongst many sources of psychosocial stress, but then does not make a clear case why this is an insufficient principle to account for the extant evidence. (I am quite willing to believe that it might be, but you need to make a clearer and more forceful argument that this is so and hence that your more esoteric model is needed). Generally, in reviewing these different approaches, you need to be much clearer which ones you are dismissing as unable to account for existing evidence, and why, and do a better job of selling the need for your model.

On a related note, when you come back to your summary at l. 518, you again imply that all previous models rely on undemonstrated assumptions about parents being good cues, but the weathering approach does not do this. Also, I am not sure that other approaches are incapable of making predictions about how father-absence effects should vary across cultures, as you imply for example at 524. The weathering hypothesis, for example, would seem to me to predict strong father-absence effects only in societies where male investment is of major importance for infant health and well-being. In societies where this is not the case, father absence should not make any difference, but something else (e.g. grandmother absence) might have a larger effect. So your approach is not unique in its ability to predict cross-cultural variation.

None of this is to detract from your very interesting model and ideas, but it's important for what you say about the prior literature to be fair and for you to persuade the reader of your unique selling points. A great thing about your model is it predicts why adoption (including evacuation) should have such strong accelerating effects, since these people grow up in households where they are completely unrelated to both parents and hence have no inclusive fitness interests in them.


Minor comments

1. There is quite a lot of unclarity in the writing, and I felt it could be edited for both conciseness and clarification.
e.g. line 102. 'However, this model...' Which model? You have described about half a dozen! NB some actually are models, but most are theories (or hand-waves at theories).
e.g. on line 139. 'We describe a more general framework...' more general than what? Why not just say general?

2. Line 212: I get what the c parameter represents, but it is a little confusing to describe it as paternity uncertainty. Paternity uncertainty is usually taken to be an attribute of males referring to their probability of actually being the sire of their partner's offspring. Here, c is in fact the probability of a marriage (in the older generation) continuing from the first to the second child. The older woman could have two offspring by different fathers, the respective fathers could be entirely sure about which one of the offspring was theirs, and yet c would be zero. The two males would be under no uncertainty. I would instead describe c as the 'paternal continuity probability' to distinguish it from the more usual usage of paternity certainty. This is particularly important since at line 425 you use paternity uncertainty in the more familiar sense.

3. Line 410: This is a very loose use of 'honest signal' (the most overused phrase in behavioural ecology). I think you just mean valid cue. Read your Maynard Smith!

4. Line 462. I would have liked you to spell out a more explicit prediction here, rather than leaving it to the reader to fill it in. e.g. 'Therefore, we predict that parental absence effects on reproductive timing will be stronger in blah blah blah...'. All of the 4 predictions in this section could do with being more explicit in this way.

5. Line 35. developping typo

6. Line 114. I felt that Van den Berg et al's (2013) paper in E&HB needed some discussion in here.

·

Basic reporting

The paper provides a nice overview of previous theory on within-family conflict and how this is resolved. This topic has been of central interest to researchers working on cooperatively breeding insects and vertebrates, and much of the evolutionary theory was developed with these systems in mind.

Experimental design

The models are simple, well explained and the analysis is logical. The results are helpful in elucidating the conditions for which mothers and daughters agree or conflict about production of offspring. I would disagree with the claim (at line 139) that the models offer a more general framework than other models of intergenerational conflict, particularly that of Cant & Johnstone 2008 PNAS (hereafter C&J). First, the Moya and Sear model assumes female philopatry, so it is difficult to argue that it is more general than C&J, which allowed dispersal to vary continuously from female-biased to no bias and even male-biased dispersal (see for example Cant et al 2009 In Reproductive Skew in Vertebrates; Johnstone & Cant 2010 Proc B).

Second, C&J assumed that older females had an efficiency advantage in reproductive conflict, and showed that despite this younger females can win out. In the Moya and Sear model, the offspring of older females can only be equal or weaker than the offspring of younger females, so it is less surprising that this model predicts a region where young females win. But this immediately raises the question of why the pattern of younger females winning in reproductive conflict is so unusual in other cooperatively breeding vertebrates: what makes humans different? Demography can potentially provide an answer to this question, but I am not sure what the Moya/Sear model says on this question. Why, according to your model, don't younger females dominate reproduction in other species?

An assumption that limits the generality of the Moya and Sear model is that the conflict is won by the individual who stands to gain more from being the first mover. This is an abstract way to model the resolution of conflict, and would lead to a competition among females to give birth first. In sequential (or Stackelberg) models in which there is a first mover advantage, the expected outcome is the same as a simultaneous solution in which each player has no advance information about the action of the other (see Cant & Shen-Feng 2006 Proc B). There are circumstances where social conflict can lead to a stable sequential order of play in which both players prefer to act in sequence (rather than simultaneously) and both agree on who should go first - in these cases the competition is said to exhibit 'endogenous timing'. I don't think the models of Moya and Sear exhibit endogenous timing but it would be interesting to examine this question.

These are quite technical points but my more broader point is that it is important to think about how abstract theoretical concepts relate to reality. In this case I think the paper would be improved if the authors did more to explain the biological meaning of the solution concept they adopt.

Validity of the findings

The findings all appear valid, given the assumptions of the model.

Additional comments

The references to the cooperative breeding literature are very welcome and highlight the commonality between the human case and other cooperatively breeding vertebrates. I also welcome the references to Reeve & Keller (1995) and Emlen (1995) since these authors really did work to highlight intergenerational conflict within families and how it might be expected to play out.. However, contrary to line 61 Reeve & Keller (1995) didn't really model the resolution of conflict in favour of parents' versus offsprings' reproduction - they solved for the differing staying incentives in mother-daughter versus sibling associations on the assumption that a single dominant female had full control over the partitioning of reproduction. If you want to cite models that do examine the resolution of conflict then Reeve et al 1998 (the tug-of-war model) would be better; and C&J was the first model (to my knowledge) that explicitly looked at the resolution of intergenerational reproductive conflict under a wide set of assumptions (e.g. sequential versus simultaneous play, varying dispersal rates).

---

## Round 0.2 · accepted · Accept

Thank you for your thoughtful response to the reviewers' feedback. I am confident that both reviewers are now satisfied with the revisions that you've made.

·

Basic reporting

No comments

Experimental design

N/A

Validity of the findings

No comments

Additional comments

Thanks for your careful revisions - it's a really interesting paper.